# Early Splicing Complexes and Human Disease

**DOI:** 10.3390/ijms241411412

**Published:** 2023-07-13

**Authors:** Chloe K. Nagasawa, Mariano A. Garcia-Blanco

**Affiliations:** 1Human Pathophysiology and Translational Medicine Program, Institute for Translational Sciences, University of Texas Medical Branch, Galveston, TX 77555-5302, USA; qsk9bk@virginia.edu; 2Department of Biochemistry and Molecular Biology, University of Texas Medical Branch, Galveston, TX 77555-5302, USA; 3Department of Microbiology, Immunology and Cancer Biology, University of Virginia, Charlottesville, VA 22903-2628, USA; 4Institute of Human Infections and Immunity, University of Texas Medical Branch, Galveston, TX 77555-5302, USA; 5Department of Internal Medicine, University of Texas Medical Branch, Galveston, TX 77555-5302, USA

**Keywords:** splicing, commitment complex, pre-spliceosome complex, cancer, Alzheimer’s diseases, multiple sclerosis

## Abstract

Over the last decade, our understanding of spliceosome structure and function has significantly improved, refining the study of the impact of dysregulated splicing on human disease. As a result, targeted splicing therapeutics have been developed, treating various diseases including spinal muscular atrophy and Duchenne muscular dystrophy. These advancements are very promising and emphasize the critical role of proper splicing in maintaining human health. Herein, we provide an overview of the current information on the composition and assembly of early splicing complexes—commitment complex and pre-spliceosome—and their association with human disease.

## 1. Introduction

Pre-mRNA splicing is a process that involves the removal of introns from the transcript and the subsequent ligation of flanking exons to generate a mature coding mRNA. In fact, the same splicing process is operative in non-coding RNAs transcribed by RNA polymerase II. Splicing is carried out by one of two multi-megadalton ribonucleoprotein complexes called the major and the minor spliceosomes [1]. The major spliceosome, which is the focus of this review, consists of five small nuclear ribonucleoprotein complexes (snRNPs): U1, U2, U4, U5 and U6, and hundreds of proteins. Each snRNP is composed of uridine-rich small nuclear RNAs (U snRNAs) and a set of associated proteins [2,3,4]. The assembly of these components onto the pre-mRNA is a highly coordinated process.

In brief, the initial complex to form is the commitment complex, which identifies the junctions between exons and introns through a series of RNA–protein and protein–protein interactions mediated by the U1 snRNP and several proteins. The process of defining these junctions has been described by two concepts: intron definition and exon definition (reviewed by de Conti et al. [5]). Intron definition is more common in pre-mRNAs with short introns and longer exons, such as those found in the yeast *S. cerevisiae*. Conversely, exon definition is more prevalent in mammalian pre-mRNAs, in which the introns are usually long (average = 3.4 kilobases) and exons are much shorter (average = 145 bases) [6,7]. Regardless of whether formed by intron or exon definition, the commitment complex is the substrate for formation of the second stable splicing complex called the pre-spliceosome complex, which promotes the interaction between the U1 and U2 snRNPs. The pre-spliceosome complex seeds the formation of the spliceosome that becomes activated to allow for catalysis via two transesterification reactions [8]. Ultimately, these reactions lead to the production of an intron lariat and a spliced product with ligated exons.

Pre-mRNA transcripts contain sequence motifs that direct the RNA–protein and protein–protein interactions required for spliceosome assembly. Therefore, any sequence alteration to the pre-mRNA transcript can misdirect the spliceosome and result in dysregulated splicing. Moreover, perturbations to the components of the spliceosome, whether through genetic variants/mutations or aberrant expression, can also cause mis-splicing. Given the importance of pre-mRNA splicing in gene expression, any alterations to this mechanism can profoundly impact the development of human disease [9].

In this review, we detail the molecular mechanisms involved in the assembly of the commitment and pre-spliceosome complexes. We highlight the critical pre-mRNA sequences that promote these early steps of spliceosome assembly, as well as the RNA-protein and protein–protein interactions that recognize and interact with these sequences to generate functional complexes. Furthermore, we provide examples that illustrate how alterations to these early splicing components can impact spliceosome formation and function and discuss the significance of these alterations in the context of human disease.

## 2. Assembly and Function of Early Splicing Complexes

### 2.1. Commitment Complex

Many fundamental discoveries regarding spliceosome assembly and function were initially made in yeast and later applied to elucidate the assembly and function of human spliceosomes (reviewed by Wilkinson et al. [10]). The first stable complex discovered in splicing competent *S. cerevisiae* extracts was the commitment complex (CC) [11]. The CC is also referred to as the early (E) complex in mammalian systems [12]. In this review, we adopt the nomenclature used in yeast and refer to the first splicing complex formed in humans as CC, based on functional template commitment assays [12,13].

Pre-mRNA transcripts contain sequence motifs that guide their interaction with spliceosomal RNAs and proteins. In this review, we present these motifs from an intron-centric perspective; however, it has been evident since Berget and colleagues proposed the exon definition concept that these sequences are also recognized across exons [14]. The 5′ end of the intron and adjacent exonic sequences consists of a 5′ splice site, which is recognized by the U1 snRNP [15]. At the 3′ end of the intron, three motifs—the branchpoint sequence, the polypyrimidine tract, and the 3′ splice site—are recognized by splicing factor 1 (SF1), U2 small nuclear RNA auxiliary factor 2 (U2AF2, also known as U2AF65) and U2 small nuclear RNA auxiliary factor 1 (U2AF1, also known as U2AF35), respectively [16,17,18] (see Figure 1). All of these interactions occur independently of ATP hydrolysis and contribute to the formation of the CC [19]. The molecular mechanism of each interaction is discussed in greater detail below.

#### 2.1.1. Binding of U1 snRNP to the 5′ Splice Site

The U1 snRNP, composed of U1 snRNA and associated proteins, binds to the 5′ splice site (SS) located at the upstream exon–intron junction (Figure 1). The consensus 5′SS motif in mammals is typically CAG|guragu, where the “|” symbol denotes the exon–intron junction and r represents a purine [22]. While there can be variations in the 5′SS sequence that may affect its interaction with U1 snRNP, there is a high degree of conservation of g and u at positions +1 and +2 of the intron, respectively. The 5′SS motif base pairs with the 5′ end of the U1 snRNA. The nucleotides A and C at positions +7 and +8 of the U1 snRNA are crucial for the formation of the RNA duplex between the 5′SS of the pre-mRNA and U1 snRNA [20] (Figure 1). Moreover, the closer the 5′SS is to consensus, the higher the affinity for the U1 snRNA [23]. It follows that the binding of U1 snRNA to a 5′SS with low affinity is more dependent on the recruitment of additional factors [23].

The RNA duplex formed by the 5′SS and U1 snRNA is stabilized by three U1-specific proteins (U1A, U1C and U1-70K). The U1 snRNA consists of four stem-loop structures that interact with spliceosomal and accessory proteins to stabilize complex formation and recruit additional factors (Figure 1). Stem-loop 1 (SLI) binds to U1-70K through its RNA recognition motif (RRM) located at its N-terminus [21]. This interaction is facilitated by the binding of phosphorylated serine/arginine splicing factor 1 (SRSF1) to the pre-mRNA [24,25]. The RNA duplex formed between the 5′SS and 5′ end of U1 snRNA is further stabilized through an interaction with the zinc-finger domain of U1C, which is recruited to the complex via U1-70K [20,21] (Figure 1). Additionally, the N-terminal RRM of U1A binds the SLII [26,27]. Although SLIII and SLIV do not directly bind U1-specific proteins, studies have shown that they interact with the DEAD-Box Polypeptide 39B (DDX39B, also referred to as U2AF65-associated protein 56 (UAP56)) and splicing factor 3a subunit 1 (SF3A1), respectively [28,29]. These interactions facilitate the transition from CC to pre-spliceosome complex formation and subsequently contribute to pre-spliceosome complex stabilization [29]. Additionally, the U1 snRNP contains seven Sm proteins (SmB/SmB′, SmD1, SmD2, SmD3, SmE, SmF and SmG) that assemble into a Sm ring [4,30]. The Sm ring interacts with the Sm site located at the 3′ end of the U1 snRNA and facilitates snRNP maturation [31]. Ultimately, recognition and binding of the U1 snRNP to the 5′SS facilitate the assembly of additional splicing factors, such as SR proteins, onto the pre-mRNA.

#### 2.1.2. Binding of SF1 and the U2AF Heterodimer to the Branchpoint Sequence, Polypyrimidine Tract and 3′ Splice Site

The BPS in yeast is a highly conserved uacuaac motif, whereas, in humans, the BPS is more variable. The BPS motif in humans is yuray, where y are pyrimidines, r is a purine and the underlined a is the branchpoint adenosine [32,33]. Despite this high degree of degeneracy, human SF1 is able to identify the BPS with help from additional protein interactors, including U2AF2 [16]. SF1 contains two types of RNA-binding motifs, a hnRNP K homology (KH) domain and a Zinc (Zn) finger domain. The KH domain of SF1 is both necessary and sufficient for the recognition of the BPS. More specifically, hydrogen bonding between the branchpoint adenosine and the amino acid isoleucine at position 177 (I177) of the KH domain creates a pocket for the adenosine [34,35] (Figure 1). This adenosine will bulge out during pre-spliceosome formation and direct the formation of the intron lariat in later steps of splicing [33]. The Zn finger domain of SF1 interacts nonspecifically with the RNA to stabilize its binding to the BPS [34]. The N-terminus of SF1 contains a domain called the U2AF-homology ligand motif (ULM) that is responsible for its binding to U2AF2 [16,36].

The U2AF complex is composed of two proteins, U2AF2 and U2AF1, the former being the larger of the two. U2AF2 interacts with the polypyrimidine (py) tract at the 3′ end of the intron (Figure 1). Py tracts are sequences composed of C and U nucleotides, and their strength as a splicing *cis*-element depends on the number of consecutive Us present. A weak py tract consists of four consecutive Us while a strong py tract consists of nine or more consecutive Us [37,38]. Two independent studies have demonstrated that substitution of Us with Cs in the py tract of a splicing substrate or chemical modifications to uridines of the py tract severely reduced its ability to splice and its affinity for U2AF2, respectively [39,40]. More recently we have shown that splicing of certain introns with C-rich and U-poor py tracts, such as those found in the *FOXP3* gene, requires high levels of DDX39B, which we will elaborate on later [41].

In addition to the U2AF homology motif (UHM) of U2AF2 that interacts with the ULM of SF1, U2AF2 has three domains that are critical for its interaction with the pre-mRNA. U2AF2 possesses two RRM domains that are responsible for binding to nine consecutive nucleotides of the py tract [42] (Figure 1). A study conducted by Mackereth et al. examined the binding affinity of U2AF2 for RNA substrates with varying amounts of consecutive Us interrupted by four to eight A nucleotides. Their results showed that while the binding of RRM2 of U2AF2 was similar across the different substrates, the overall binding affinity of U2AF2 varied. This discrepancy in binding affinity was attributed to RRM1, where the highest affinity occurred with an RNA substrate consisting of nine consecutive Us [38]. The binding affinity differences of U2AF2 for the RNA substrates can be attributed, in part, to the different conformations of the RRM domains. The two RRM domains adopt a closed, ‘inactive’ conformation in the absence of RNA substrate or in the presence of a weak py tract, or an open, ‘active’ conformation in the presence of a strong U-rich py tract [43].

Since the py tract is composed of both C and U nucleotides, the Kielkopf group conducted additional mutational analyses that demonstrated the RRM2 of U2AF2 is more sensitive to the presence of Cs in the py tract than RRM1 [44]. The conformational changes of U2AF2 potentially explain why weak py tracts composed of C residues have a decreased affinity for U2AF2, as it adopts a closed conformation, resulting in less efficient splicing compared to their U-rich py tract counterparts. The third domain, the ULM at the N-terminus of U2AF2, binds to the UHM of the smaller U2AF subunit, U2AF1 [45].

U2AF1 binds the 3′ splice site (3′SS) at the intron–exon junction. The consensus 3′SS motif is typically yag|GU, where y represents a pyrimidine and “|” denotes the intron–exon junction. The a and g nucleotides at positions -2 and -1 of the intron are conserved in all introns spliced by the major spliceosome [46] (Figure 1). U2AF1 has an RRM domain that is flanked by two Zn finger domains. Yoshida et al. demonstrated that the RRM domain does not directly interact with the RNA but instead serves as a scaffold for the Zn finger domains to cooperatively bind to the 3′SS [47]. Moreover, U2AF1 binds to three nucleotides upstream and twelve nucleotides downstream of the intron–exon junction, enhancing the binding and stability of the complex [46].

There are additional interactions that can influence the formation of stable CC, including the role of serine/arginine (SR) proteins. SR proteins are RNA-binding proteins that help regulate splicing [48]. Studies have demonstrated important interactions between SR proteins and U1-70K in stabilizing early splicing complexes [49,50]. Furthermore, our group reconstituted CC from purified human components and showed a requirement for phosphorylated serine/arginine-rich splicing factor 1 (SRSF1, previously known as ASF/SF2) to form stable CC (Jamison and Garcia-Blanco, unpublished results). Although additional splicing factors, such as SRSF1, were not discussed in detail above, it is important to note that they play critical roles in stable CC formation. Nevertheless, the assembly of stable CC serves as the substrate for the subsequent formation of the pre-spliceosome complex.

### 2.2. Pre-Spliceosome Complex

The pre-spliceosome, also referred to as the A complex, is the next splicing intermediate to assemble (Figure 2A). ATP binding and hydrolysis by two DExD-box helicases—DDX39B/UAP56 (or its paralog DDX39A) and DDX46—allows for the U2 snRNP to interact with the pre-mRNA and form the pre-spliceosome [51,52,53].

#### 2.2.1. Displacement of SF1 from the Branchpoint Sequence

DDX39B/UAP56 is an ATP-dependent DExD-box helicase that was shown to interact with residues 138–183 of U2AF2 and demonstrated to be essential for the recruitment of the U2 snRNP to the branchpoint sequence [51,52]. The Green lab showed that human nuclear extracts depleted of DDX39B were unable to splice an RNA substrate and exhibited poor formation of pre-spliceosome complexes [51]. Interestingly, DDX39B has a paralog named DDX39A (also known as UAP56-related helicase 49kDa (URH49)). While DDX39B and DDX39A are ~90% identical proteins and have some redundant functions, studies have shown that they have distinct roles in cellular functions including mRNA transport [54] and alternative splicing [55]. It is unclear, however, if the in vitro splicing studies that defined UAP56 activity could distinguish between DDX39B and DDX39A, and our assumption is that the two paralogs contributed to the activity observed. Our lab is currently working on distinguishing their roles in pre-mRNA splicing.

Investigations by the Guthrie lab revealed data suggesting that the ATPase activity of the yeast homolog of DDX39B, Sub2, is crucial for the displacement of SF1 from the BPS. The displacement of SF1 allows for the recruitment and binding of the U2 snRNP to the BPS [56] (Figure 2B).

#### 2.2.2. Recruitment and Binding of U2 snRNP to the Pre-mRNA

Upon release of SF1, the branchpoint is accessible to interact with the U2 snRNP. The U2 snRNP consists of a 188-nucleotide U2 snRNA and a group of proteins that form the SF3A and SF3B complexes, as well as the Sm ring and accessory proteins U2A′ and U2B′ [2,57]. The U2 snRNA is composed of five domains: stem-loop I, branch point recognition sequence, stem-loop II, Sm site and the 3′ end comprising stem-loops III and IV [58,59]. The Sm ring, U2A′ and U2B′ bind to the 3′ end of the U2 snRNA and remain stably bound during pre-spliceosome formation. These proteins play important roles in stabilizing the U2 snRNP structure [60]. In contrast, the proteins that form the SF3A and SF3B complexes do not remain stably bound to the U2 snRNA. Their roles are to bind to and remodel the interaction between the pre-mRNA and the pre-spliceosome [61]. The SF3A complex is composed of three proteins: SF3A1-3 (Prp21, Prp11 and Prp9 in yeast, respectively). The SF3B complex consists of seven proteins: SF3B1-7 and all but SF3B6 have a yeast homolog (Hsh155, Cus1, Rse1, Hsh49, Ysf3 and Rds3, respectively) [4]. The explanation behind the absence of a yeast homolog for human SF3B6 is addressed below.

The branch point recognition sequence of U2 snRNA mediates an RNA–RNA interaction between the U2 snRNA and the BPS of the intron. The branch point recognition sequence contains a consensus sequence 5′-GΨAGUA-3′ (where Ψ is pseudouridine) that binds to the BPS 5′-yuray-3′, forming an RNA duplex known as the branch helix [62] (Figure 2C). The branch point adenosine is bulged out into a pocket formed by proteins of the SF3B complex, mainly SF3B1 and SF3B7 (also known as PHF5A) [62]. Studies by the Reed group demonstrated that SF3B1 interacts with the upstream and downstream nucleotides of the branchpoint, establishing an interaction with the UHM domain of U2AF2 and stabilizing the RNA–protein interaction [63]. The BPS in humans is highly degenerate compared to its yeast counterpart, possibly explaining the absence of a yeast homolog for SF3B6 (also known as p14). SF3B6 binds the 5′ end of the branch helix, stabilizing the RNA duplex, especially for branchpoint sequences with poor complementarity to the branch point recognition sequence [64]. During pre-spliceosome formation, additional RNA–protein contacts are established. Nucleotides upstream of the branchpoint pass through a tunnel of SF3 proteins—SF3A2, SF3A3, SF3B2 and SF3B4. Stem-loop I interacts with SF3B4 and SF3A3, while the 3′ end of U2 snRNA binds to U2A′ and U2B′ that interact with the U1 snRNP (reviewed in van der Feltz and Hoskins [65]).

Along with RNA–protein interactions, protein–protein interactions are critical for the proper assembly of the pre-spliceosome. SF3B1 contains twenty Huntingtin, elongation factor 3, the A subunit of protein phosphatase 2A and signaling kinase TOR1 (HEAT) repeats. These repeats form a ring conformation around the branch helix and can adopt a closed or open conformation depending on interactions with other proteins. Two proteins, DDX46 (Prp5 in yeast) and HIV-1 Tat specific factor 1 (HTATSF1, Cus2 in yeast), facilitate the conformational changes in SF3B1 [53,62,66]. HTATSF1 was first identified biochemically as a human host cofactor involved in Tat-directed HIV-1 transcription [67] but was later demonstrated to be primarily acting on the splicing of HIV-1 transcripts [68]. More recently, HTATSF1 was shown to immunoprecipitate with the U2 snRNP [69]. The Galej group revealed that HTATSF1 binds to both SF3B1 and the U2 snRNA, preventing premature closure and incorrect recognition of the BPS [62]. In an ATP-dependent manner, DDX46 displaces HTATSF1 to allow for the branch point recognition sequence of U2 snRNA to probe for the BPS. Upon correct binding of the branch point recognition sequence with the BPS, the branchpoint adenosine bulges out into the pocket formed by SF3B1 and SF3B7. This interaction allows for SF3B1 to adopt a half-closed conformation [53,62]. DDX46 also physically interacts with proteins of the SF3A and SF3B complexes to hinder the complete closing of SF3B1 until branch helix formation. Upon branch helix formation, SF3B1 transitions to a closed conformation, displacing DDX46 and allowing for additional conformational changes within the U2 snRNP to complete pre-spliceosome assembly [53].

The accurate identification of the 5′SS, BPS, py tract and 3′SS by snRNPs and associated proteins is the backbone of proper splicing. In humans, genetic variants or single nucleotide polymorphisms (SNPs) can create or disrupt these consensus motifs, leading to alterations in splicing patterns, gene expression and, ultimately, disease susceptibility. Furthermore, any perturbation in the expression or function of spliceosomal components can alter normal splicing and have major implications in human disease. In this review, we focus on how various alterations to the assembly of commitment and pre-spliceosome complexes lead to disease.

## 3. Alterations to the Formation of Early Splicing Complexes Leads to Human Disease

Due to the critical role of splicing in gene regulation, any disruption to the spliceosome itself can have substantial consequences on the transcriptional and translational landscape, with significant implications in human disease. The expanding knowledge of spliceosome assembly and function has elucidated the molecular mechanisms that underly splicing-related diseases, including various cancers, neurodegenerative diseases, autoimmune diseases and developmental defects. Here, we review alterations to components of the commitment and pre-spliceosome complexes and their associated diseases (summarized in Table 1).

### 3.1. U1 snRNP

As described previously, the U1 snRNP is critical for 5′SS recognition during commitment complex assembly. A study conducted by the Lah group revealed cytoplasmic aggregates of U1 snRNA via RT-qPCR and RNA hybridization assays in post mortem human brain and spinal cord samples from both sporadic and familial cases of Alzheimer’s disease (AD) [70]. Although the role of U1 snRNA aggregation in AD pathogenesis has not been fully elucidated, it is reasonable to posit that the aggregation of the U1 snRNA in the cytoplasm reduces levels of functional U1 snRNA and contributes to disease progression through its effect on RNA processing [70]. In addition to levels of U1 snRNA impacting human disease, mutations in *U1 snRNA*, encoded by four U1 snRNA genes and three pseudogenes [85], have been associated with the development of cancers. In chronic lymphocytic leukemia and hepatocellular carcinoma, there is an increased frequency of A to C somatic mutations at the third position of the *U1 snRNA*. This mutation changes the recognized 5′SS sequence from an ag to a cg, creating new splice junctions that alter the splicing of genes, including those that promote tumorigenesis [71]. A separate study revealed the presence of an A to G hotspot mutation at the third position of the *U1 snRNA* in approximately 50% of Sonic-hedgehog-driven medulloblastomas. This mutation also leads to cryptic 5′SS usage and specifically inactivates tumor suppressor genes while activating oncogenes [72].

Alterations in U1-70K, a protein component of the U1 snRNP, have been implicated in the development of neurodegenerative disorders. Cleavage at the N-terminus of the protein produces an N-terminal, 40 kDa fragment of U1-70K that is referred to as N40K [73]. Although the exact mechanism of cleavage is not specified, a study conducted by Chen et al. revealed that N40K exerts a dominant-negative effect on wild-type U1-70K, affecting its ability to assemble into the commitment complex [73]. In both an AD mouse model and primary human neurons, this led to splicing defects in genes involved in synaptic signaling and resulted in GABAergic synapse deregulation and neuronal hyperexcitability [73]. Moreover, alternative splicing of U1-70K has been suggested to play a role in the pathogenesis of amyotrophic lateral sclerosis (ALS), specifically the subtype associated with oxidative stress (ALS-Ox) [86]. The alternatively spliced product introduces an alternative exon between exons 7 and 8, resulting in a shorter U1-70K product that is more prevalent in patients with ALS-Ox [74].

### 3.2. U2AF Heterodimer

The U2AF heterodimer is composed of U2AF1 and U2AF2. Alterations to these two proteins have been known to contribute to human disease.

S34F/Y substitution in the Zinc finger 1 domain of U2AF1 has been detected in approximately 12% of patients with myelodysplastic syndromes (MDSs), a group of disorders characterized by the abnormal formation and function of blood cells [87,88]. This mutation has been demonstrated to alter splicing in different cell types by promoting exon skipping [75,76]. Additionally, the S34F mutation has been shown to affect pre-mRNA processing of the autophagy-related factor 7 gene (*ATG7*), resulting in decreased expression and function of ATG7, predisposing cells to oncogenesis [77]. A less frequent mutation observed in patients with MDS affects the second Zinc finger domain of U2AF1. This involves an A to G missense mutation at position 470 of the coding sequence in exon 6, resulting in a Q157R substitution. Furthermore, this missense mutation generates an alternative 5′SS in exon 6, leading to the deletion of four amino acids (ΔYEMG) downstream of the Q157R substitution [78]. Since this mutation occurs in the second Zn finger domain of U2AF1, which is responsible for binding to the 3′SS, the shortened U2AF1ΔYEMG protein recognizes cryptic 3′SS and leads to mis-splicing of genes associated with oncogenesis including *DICER1* [78].

Although less characterized, various cancers have been associated with alterations to U2AF2. An SNP in *U2AF2*, *rs310445*, has been associated with a risk of pancreatic cancer. Specifically, individuals homozygous for the risk allele (C) had a 1.31-fold higher likelihood of developing pancreatic cancer when compared to those homozygous for the non-risk allele (T) [79]. In 2020, the Kielkopf group published a study investigating the effects of two substitutions in the RRM1 and RRM2 domains of U2AF2, N196K and G301D, respectively, on its RNA-binding ability [80]. The N196K substitution is associated with acute myeloid leukemia and stabilizes the open conformation of U2AF2, thereby increasing its RNA-binding affinity. The G301D substitution is associated with colon adenocarcinomas and castration-resistant prostate carcinoma and does not induce a conformation change to U2AF2 but does impact RNA-binding affinity. Although the exact mechanism of how these mutations contribute to disease pathogenesis is unclear, it is possible that the stabilized open conformation of the N196K substitution could recognize and bind weak polypyrimidine tracts and potentially introduce cryptic splice site usage. On the contrary, the G301D substitution could lead to the production of shortened non-functional or completely absent isoforms due to decreased RNA affinity. Therefore, both of these substitutions could differentially influence splicing and gene expression, potentially playing different roles in disease pathogenesis [80].

### 3.3. DDX39B

DDX39B/UAP56 plays critical roles in spliceosome assembly. Above we described the role of its ATPase domain in displacing SF1 from the BPS, allowing for U2 snRNP binding during pre-spliceosome complex formation. However, it is unclear whether these studies distinguished between DDX39B and its highly homologous paralog, DDX39A. A study conducted by Nakata et al. investigated the role of DDX39A and DDX39B in the splicing of a variant of the androgen receptor (AR) called AR-V7 [89]. The AR-V7 transcript results in a ligand-independent protein isoform that contributes to the growth of castration-resistant prostate carcinoma and correlates with poor prognosis. Nakata et al. demonstrated that the depletion of both DDX39A and DDX39B led to the decreased production of the AR-V7 transcript, as well as the decreased expression of a downstream target of AR-V7 called kallikrein-related peptidase 3 (KLK3) [89]. This suggests that both DDX39A and DDX39B are involved in the splicing of the AR-V7 transcript and have implications for the development of castration-resistant prostate carcinoma. While this study implies some redundancy between DDX39A and DDX39B, other groups have demonstrated that these paralogs also possess non-overlapping functions. Our lab is working on distinguishing the roles of DDX39A and DDX39B in splicing and their contributions to autoimmunity (Banerjee and Garcia-Blanco, unpublished results).

In 2017, our group showed that DDX39B was a trans-acting factor that activated the splicing of exon 6 of interleukin-7 receptor (*IL7R*), a gene critical for T cell function and homeostasis [55,90]. Exon 6 encodes for the transmembrane domain of IL7R; therefore, its inclusion leads to a transmembrane-bound isoform, while skipping of exon 6 results in a soluble isoform of IL7R (sIL7R), which is involved in the pathogenesis of an autoimmune disease called multiple sclerosis (MS) [91]. We demonstrated that CD4^+^ T cells depleted of DDX39B had increased skipping of exon 6 and upregulated production of sIL7R. Furthermore, we identified an SNP, *rs2523506*, in the 5′ untranslated region (UTR) of *DDX39B/UAP56* that was associated with an increased risk of MS [55]. We demonstrated that individuals homozygous for the A risk allele had a 50% decrease in protein expression, while heterozygous individuals had a 25% decrease compared to those homozygous for the C non-risk allele. Moreover, we showed epistasis between rs2523506 in *DDX39B* and an SNP in *IL7R*, rs6897932, that is strongly associated with MS [92,93,94]. Using a logistic regression model, we measured the joint effect of SNPs rs2523506 in *DDX39B* and rs6897932 in *IL7R* on MS risk. The population that was homozygous for both risk alleles, A in rs2523506 and C in rs6897932, had approximately a threefold increased risk of MS compared to the population homozygous for the non-risk alleles in both genes [55]. The increased risk of MS in the population homozygous for *DDX39B* and *IL7R* risk alleles is likely due, in part, to the necessity of DDX39B for the activation of IL7R exon 6 splicing.

More recently, our group published on the role of DDX39B in the splicing of *FOXP3* transcripts [41]. FOXP3 is critical for the development, maintenance and function of T regulatory (Treg) cells and has implications in autoimmune disorders [95]. We demonstrated that in CD4^+^ T cells depleted of DDX39B, there was a decrease in FOXP3 RNA and protein expression. The decrease in FOXP3 expression was caused by increased retention of *FOXP3* introns, which have weak C-rich py tracts. Moreover, the depletion of DDX39B led to decreased IL2RA (CD25) expression, which is critical for Treg cell function. Together, these data illustrate that DDX39B expression is critical for FOXP3 expression and function. Therefore, DDX39B expression is essential for immunoregulatory gene expression and function, and its decreased expression impacts the development of autoimmune diseases.

### 3.4. U2 snRNP

The U2 snRNP plays a critical role in forming the pre-spliceosome complex, which serves as the foundation for the subsequent assembly of the spliceosome. A study conducted by Tian et al. revealed a relationship between an SNP, *rs2074733*, in *SF3A1* and pancreatic cancer. The population homozygous for the risk allele (C) had an increased risk of developing pancreatic cancer compared to the population heterozygous or homozygous for the non-risk allele (T) [81].

SF3B1 is the most commonly mutated spliceosomal gene in patients with MDS, accounting for approximately 30% of cases [96,97,98]. The frequency of *SF3B1* mutations is significant and has been included as a diagnostic criterion of MDS with ring sideroblasts, which are immature erythroid cells (MDS-RSs). Patients are diagnosed with MDS-RS if 15% or more of their nucleated erythroid cells are RS, or if 5% or more are RS and they have a somatic mutation in *SF3B1*, with 83% of patients displaying such a mutation [88,97,98,99]. Furthermore, SF3B1 is mutated in 10–15% of patients with chronic lymphocytic leukemia [100], highlighting the importance of understanding the role of SF3B1 in disease pathogenesis. SF3B1 mutations typically result in aberrant splicing due to the recognition and utilization of cryptic 3′SS. Many of these critically spliced transcripts undergo nonsense-mediated decay affecting gene expression [101,102]. One specific substitution in SF3B1, K700E, found in patients with MDS-RS, has been demonstrated to disrupt its interaction with the SURP and G-patch domain containing 1 (SUGP1) protein. SUGP1 is a splicing factor that activates DEAH-box RNA helicases and functions to recognize branchpoints. The disruption of the SF3B1 and SUGP1 interaction consequently affects branchpoint recognition, leading to the utilization of a cryptic 3′SS located 10–30 nucleotides upstream of the canonical 3′SS [82]. Lastly, increased SF3B1 RNA and protein expression in patients with hepatocellular carcinoma is associated with worse disease outcomes [103].

Although SF3B1 has been the primary focus of many U2 snRNP splicing-related diseases, there have been studies that revealed the impact of haploinsufficiency of *SF3B2* and *SF3B4* on the development of craniofacial microsomia and Nager syndrome, respectively. Timberlake et al. identified seven unrelated families with the loss of function variants in *SF3B2* that led to haploinsufficiency [83]. These patients presented with craniofacial defects, including external ear malformations and mandibular hypoplasia. Based on the knowledge of the development of the external ear and mandible, the researchers believe that SF3B2 haploinsufficiency affects the development of pharyngeal arch I. The precise changes in splicing caused by *SF3B2* haploinsufficiency are unknown, but researchers suggest that the retention of “poison exons” results in aberrant splicing in neural crest precursors and leads to disease progression [83]. A previous study by Bernier et al. revealed that 60% of patients with Nager syndrome, a rare autosomal dominant genetic disease characterized by facial abnormalities and preaxial limb malformations [104], had mutations in *SF3B4*, which were predicted to produce nonfunctional proteins [84]. The researchers proposed that the haploinsufficiency of *SF3B4* played a role in the pathogenesis of Nager syndrome, although the specific splicing mechanism is still being investigated [84].

## 4. Conclusions

Here, we discussed the early molecular interactions between consensus splice site motifs, namely, the 5′SS, BPS, py tract and 3′SS, and their associated splicing factors. These interactions play a crucial role in the assembly of the commitment and pre-spliceosome complexes, which are essential for proper spliceosome function. Disruptions in the assembly of these complexes can result in aberrant splicing events, which have implications for the development of human disease.

Advances in our understanding of spliceosome assembly and function have provided valuable insights into the mechanisms that underly splicing-related diseases. This knowledge has paved the way for the development of splicing-targeted therapeutics, such as anti-sense oligonucleotides, to treat diseases like spinal muscular atrophy and Duchenne muscular dystrophy [9]. While these therapies have been revolutionary, there is still much to unravel about the causes and consequences of dysregulated splicing in human disease. With a deeper understanding, it becomes possible to develop additional splicing-targeted therapeutics to address a broader range of diseases including cancer, neurodegenerative and autoimmune diseases, developmental disorders and more.

## Figures and Tables

**Figure 1 ijms-24-11412-f001:**
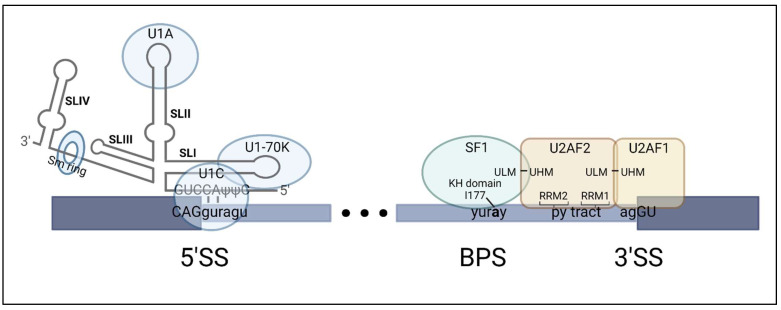
Molecular mechanism of commitment complex assembly. The U1 snRNP, which is composed of U1 snRNA, the U1 snRNP-specific proteins, U1-70K, U1A and U1C, and the Sm ring, interacts with the 5′ splice site (SS). More specifically, the 5′ end of U1 snRNA interacts with the 5′SS at the exon (dark blue)–intron (light blue) junction of the pre-mRNA. This interaction is stabilized by U1C, which is recruited to the RNA duplex via U1-70K [20,21]. U1A binds stem-loop II (SLII). At the 3′ end of the intron, SF1 binds to the branchpoint sequence (BPS), U2AF2 binds to the polypyrimidine tract (py tract) and U2AF1 binds to the 3′SS at the intron–exon junction. In particular, the isoleucine residue at position 177 of the KH domain of SF1 binds the branchpoint adenosine while the U2AF-homology ligand motif (ULM) of SF1 binds the U2AF homology motif (UHM) domain of U2AF2. U2AF2 has two RRM motifs that bind the py tract and a ULM domain that binds the UHM of U2AF1. (Created with BioRender.com (accessed on 12 May 2023)).

**Figure 2 ijms-24-11412-f002:**
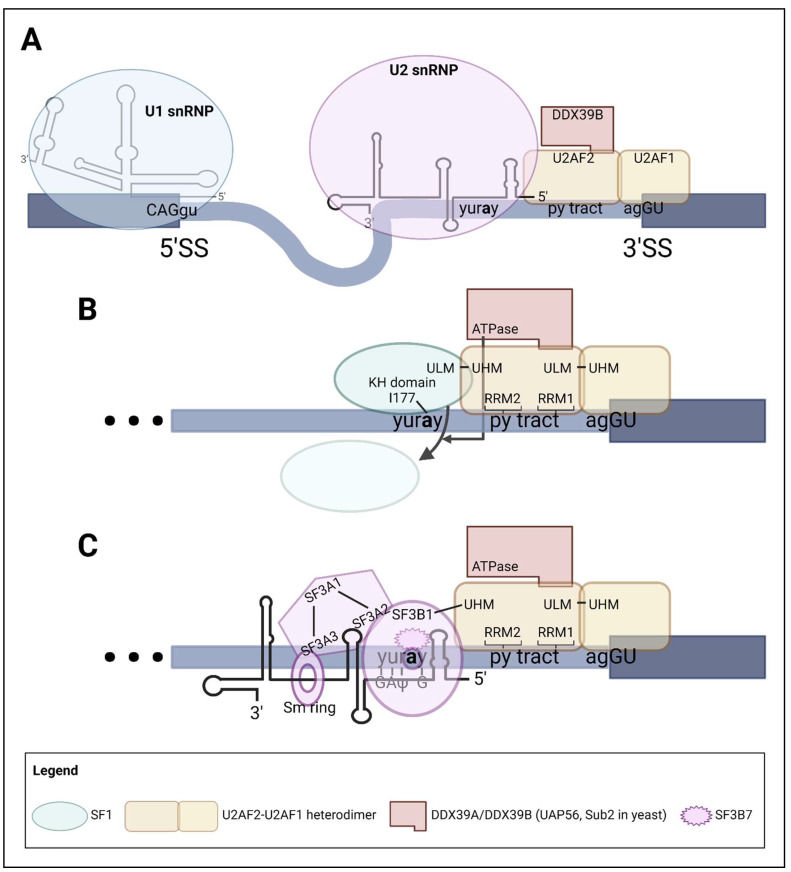
Molecular mechanism of pre-spliceosome complex assembly. (**A**) Schematic diagram of pre-spliceosome complex. The U2 snRNP associates with the pre-mRNA via binding to the BPS and its conformational changes begin to orient the pre-mRNA for ideal interaction with the U1 snRNP. (**B**) Displacement of SF1 by the ATPase activity of DDX39A/DDX39B (UAP56, Sub2 in yeast) allows for U2 snRNP to bind the branchpoint. DDX39A/DDX39B (UAP56, Sub2 in yeast) is stably bound to U2AF2. (**C**) Binding of U2 snRNP to the branchpoint sequence. The branchpoint adenosine sits in a pocket created by SF3B7 and SF3B1. SF3B1 interacts with the UHM domain of U2AF2. The SF3A complex forms a bridge-like structure between the SF3B complex and the Sm ring, stabilizing the U2 snRNP. (Created with BioRender.com (accessed on 12 May 2023)).

**Table 1 ijms-24-11412-t001:** Alterations to splicing components and their associated outcomes and disease.

Splicing Component	Alteration to Splicing Component	Outcome	Associated Disease	References
**U1 snRNP**				
U1 snRNA	Aggregation in post mortem human brain and spinal cord	RNA processing defects	Alzheimer’s disease	[70]
U1 snRNA	A→C somatic mutation at third position	Use of cryptic 5′SS	Chronic lymphocytic leukemia Hepatocellular carcinoma	[71]
U1 snRNA	A→G somatic mutation at third position	Use of cryptic 5′SS	Sonic hedgehog medulloblastoma	[72]
U1-70K	N40K isoform	Dominant-negative effect affecting the ability to assemble commitment complex	Alzheimer’s disease	[73]
U1-70K	Alternative exon between exons 7 and 8	Shorter transcript	Amyotrophic lateral sclerosis-oxidative stress (ALS-Ox)	[74]
**U2AF complex**				
U2AF1	S34F/Y substitution	Promote exon skipping	Myelodysplastic syndrome	[75,76]
U2AF1	S34F substitution	Affects pre-mRNA processing of autophagy-related factor 7	Myelodysplastic syndrome	[77]
U2AF1	Q157R substitution	Promote exon skipping	Myelodysplastic syndrome	[78]
U2AF2	SNP rs310445 (T→C)	Not specified	Pancreatic cancer	[79]
U2AF2	N196K substitution	Stabilization of open conformation; increases RNA-binding affinity	Acute myeloid leukemia	[80]
U2AF2	G301D substitution	Decreases RNA-binding affinity	Colon adenocarcinoma Castration-resistant prostate carcinoma	[80]
**DDX39B/UAP56**	SNP rs2523506 (C→T)	Decreases translation	Multiple sclerosis	[55]
**U2 snRNP**				
SF3A1	SNP rs2074733 (T→C)	Not specified	Pancreatic cancer	[81]
SF3B1	K700E substitution	Disrupt branchpoint recognition and induces usage of cryptic 3′SS	Myelodysplastic syndrome-ring sideroblasts	[82]
SF3B2	Loss of function variants	Haploinsufficiency	Craniofacial microsomia	[83]
SF3B4	Loss of function variants	Haploinsufficiency	Nager syndrome	[84]

## Data Availability

All data are presented in the manuscript and are freely accessible.

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
