# Peer review of "Early Splicing Complexes and Human Disease"

_ijms, 2023, doi:10.3390/ijms241411412_

Round 1

Reviewer 1 Report

Nagasawa and Garcia-Blanco present a detailed and well written review of the complexes that form in the early steps of pre-mRNA splicing and the diseases associated with variants/mutations within these early splicing complexes. The authors provide good, detailed, background on splicing and the first step of spliceosome assembly and the conformational changes at the 5’SS, BPS and 3’SS. This background leads in nicely to the numerous disease examples that are given in very good detail. Overall, this is a comprehensive and detailed review of the early steps of spliceosome assembly and disease that will be a valuable resource for the splicing field as well as those outside of the field.

Suggested minor corrections

Line 25 -  ..mature “coding” mRNA..

Figure 1 – do you really need a legend within this Figure? Why don’t you just add the names SF1 and U2AF2-U2AF1 heterodimer above those protein depictions or make the boxes/circles bigger on their top to accommodate the names? It may be more informative if the regions of U1 snRNA that are discussed in the text (SL1, SLII, SLIII, SLIV and Sm site) are annotated on the U1 snRNA in the Figure.

Line 124 – “additional splicing factors” is vague, can you be more specific here about what U1 snRNP binding to the 5’SS leads to?

Line 127 – The statement “The first RNA-protein interaction we detail..” is strange as the authors have just detailed numerous RNA-protein interactions in the U1 snRNP?

Line 168 - ..the RRM2 “of U2AF2” is more sensitive…

Lines 318-319 – regarding the cleavage of U1-70K to produce the N40K fragment. Is it known how or why this cleavage takes place? If it is not known, then it should be stated in the text that the mechanism by which the N40K fragment is produced is not known and but would be of interest to find out as this would be an important point for therapeutic intervention.

Line 350 - ..compared “to” those…

Author Response

We thank the reviewer for their thoughtful review and have made all requested changes

Reviewer 2 Report

In this manuscript, Nagasawa et al. discussed about early spliceosome complex and it’s misregulation in human diseases. The manuscript documented our current understanding on the structure, function and mechanistic orchestration of early spliceosome complex describing important components in the pathway, and different cases of dysregulated events. Overall, the manuscript is written in a well-organized way, with precise information, clear and easy to understandable figures and tables. I have no major concerns, but I would like to make a suggestion to authors. Although the authors mentioned in abstract “As a result, result, targeted splicing therapeutics have been developed, treating various diseases including spinal muscular atrophy and Duchenne muscular dystrophy”, however there is no detail description about therapeutic development, only one line in conclusion. If they could make a section of “Therapeutic development targeting dysregulated early spliceosome complex”, this manuscript could draw the attention of a wider range of audience. However, the suggestion is just for the consideration to authors, not absolutely required. Based on the merit of the manuscript, I recommend that it could be accepted in its present form for publication.

Author Response

We thank the reviewer for the thoughtful review and although we understand the suggestion to have a new section on therapeutics, we believe that to do justice to this would require text almost as long as the current review.